# Morphological Differentiation, Yield, and Cutting Time of *Lolium multiflorum* L. under Acid Soil Conditions in Highlands

**DOI:** 10.3390/plants13162331

**Published:** 2024-08-21

**Authors:** William Carrasco-Chilón, Marieta Cervantes-Peralta, Laura Mendoza, Yudith Muñoz-Vílchez, Carlos Quilcate, David Casanova Nuñez-Melgar, Héctor Vásquez, Wuesley Yusmein Alvarez-García

**Affiliations:** 1Dirección de Desarrollo Tecnológico Agrario, Instituto Nacional de Innovación Agraria (INIA), Estación Experimental de Baños del Inca, Jr. Wiracocha s/n, Baños del Inca, Cajamarca 06004, Peru; 2Dirección de Desarrollo Tecnológico Agrario, Instituto Nacional de Innovación Agraria (INIA), Sede Central, Av. La Molina 1981, La Molina, Lima 15024, Peru; promegnacional@inia.gob.pe (C.Q.);; 3Facultad de Ingeniería Zootecnista, Agronegocios y Biotecnología, Universidad Nacional Toribio Rodríguez de Mendoza de Amazonas (UNTRM), Cl. Higos Urco 342, Chachapoyas 01001, Peru

**Keywords:** *Lolium multiflorum* L., Cajamarquino ecotype, Andean pastures, drought tolerance, dairy farming

## Abstract

Livestock production in the basins of the northern macro-region of Peru has as its primary source pastures of *Lolium multiflorum* L. ‘Cajamarquino ecotype’ (ryegrass CE) in monoculture, or in association with white clover Ladino variety, for feeding. The objective of this research work was the morphological characterisation, yield evaluation, and cutting time evaluation of two local genotypes (LM-58 and LM-43) of *Lolium multiflorum* L. in six locations. An ANOVA was performed to compare fixed effects and interaction. It was determined that the LM-58 genotype is intermediate, growing semi-erect, with a dark green colouring and 0.8 cm broadleaf, and can reach an average stem length of 46 cm, up to 1.6 cm. day^−1^, achieving fourth-leaf growth at 28 days under appropriate management conditions. Despite the differentiated characteristics, according to BLASTn evaluation, the ITS1 sequences showed a greater than 99.9% similar identification to *Lolium multiflorum* L., characterising it as such. It was determined that the LM-58 genotype outperforms LM-43, achieving a forage yield of 4.49 Mg. ha^−1^, a seed production of 259.23 kg. ha^−1^, and an average of 13.48% crude protein (CP). The best biomass yield (49.10 Mg. ha^−1^.yr^−1^) is reached at 60 days; however, at 30 days, there is a high level of CP (14.84%) and there are no differences in the annual protein production at the cutting age of 60 and 45 days. With the results of the present study, LM-58 from a selection and crossbreeding of 680 ryegrass EC accessions emerges as an elite genotype adapted to the conditions of the northern high Andean zone of Peru.

## 1. Introduction

Grasslands constitute over 70% of the world’s agricultural land and play a pivotal role in food security, the economy, and global ecology due to their adaptability and functionality [1]. It is anticipated that climate change will intensify weather events, which will, in turn, lead to an increase in the frequency of droughts. This will harm forage production while exposing agricultural and silvopastoral systems to a heightened risk of imminent hazards [2]. Implementing measures designed to enhance genetic heterogeneity, topography, and grassland diversity can improve stress tolerance, mitigate damage, and facilitate recovery following periods of drought [3].

*Lolium multiflorum*, a grass species cultivated annually worldwide, stands out for its exceptional adaptability to diverse soil types and various climatic conditions. Its rapid establishment, high palatability, good digestibility, high yields of quality forage, and tolerance to diverse grazing methods make it a preferred choice for many farmers and producers. Its ability to reseed reduces costs and facilitates soil restoration, providing a reassuring solution in the face of climate change [4,5].

Ryegrass, in conjunction with other forage species (clover and vetch), is the primary source of feed for livestock. This is particularly true in hillside areas, where prolonged droughts impact pasture production, leading to a reversible or irreversible disruption of the plant’s functioning and structure. This, in turn, affects the economy and livelihood of the livestock community in the northern highlands of Perú [6,7,8].

*Lolium multiflorum* L. is a grass species widely used in forage production due to its rapid growth and high productivity. This plant is adapted to altitudes between 2400 and 3200 m asl and temperatures between 2 and 25 °C. In addition to exhibiting resistance to trampling, it is considered a primary source of winter forage [9]. However, in highland areas, where acid soils are prevalent, the yield of *Lolium* spp. varieties are adversely affected by low nutrient availability, yield deficit, and environmental stress [10]. The ryegrass–soil interaction is significant due to the exposure to such soils, given the grass’s high adaptability, absorption, and fixation capacity. This substantially balances crowns and roots, thereby ensuring the grass’s vigour against these challenging soils [11]. The *Lolium* family has demonstrated remarkable resistance and adaptability, enabling grasses in the northern highlands of Peru to thrive in low temperatures. This suggests that they possess a valuable source of frost and drought tolerance genes, as evidenced by their ability to activate the cold acclimatisation process [7].

Genetic improvement of *Lolium* spp. has resulted in the generation of specific genetic mutations, which have enabled the species to behave as a perennial under managed conditions due to its high competitiveness, genetic diversity, and seed production [12,13]. In recent decades, research has concentrated on enhancing yield, dry matter, and forage quality, thereby facilitating more effective crop management and enabling a more significant number of harvests per year. These productivity gains are contingent upon optimal humidity and temperature, an adequate pH level, organic matter, potassium, and phosphorus, which are essential for plant development and maturity. Nitrogen is also a crucial element, as it is vital for seedling production [14,15].

Understanding the germination behaviour of ryegrass seeds is crucial for effective integrated weed management strategies. The type of dormancy enriches seed ecology, the species’ life cycle, and seasonal changes in environmental conditions, including temperature, precipitation, and soil salinity, from dispersal to germination [16]. Ryegrass seeds initially exhibit dormancy, which may be broken and result in germination occurring at different times throughout the growing season. This knowledge empowers effective weed management strategies [17].

*Lolium multiflorum* L. is a species with a rapid growth rate, and it is characterised by its long and thin foliage with a glossy appearance and a distinctive erect and hollow stem. The root system is characterised by high fibrousness and superficiality, which enables rapid absorption of nutrients and water. In response to environmental cues, the plant initiates an adaptive process to prolong its vegetative stage, manifested in various morphological changes. These include a reduction in the height of the panicle, a decrease in the length of intact leaves, and an increase in the number of growing leaves [18].

A reduction in grazing intensity was associated with an enhanced leaf elongation rate, reaching 3.5 cm. day^−1^, and a diminished tillering density, with 150 tillers. m^−2^. Additionally, plant height, green forage yield, and dry matter ranged from 115.0 to 130.7 cm, 1407 to 3240 kg. ha^−1^, and 392 to 976 kg. ha^−1^, respectively. These results significantly impacted the attributes of ryegrass ‘CE’ (*Lolium multiflorum* L.), indicating that moderate to light grazing increases spatial heterogeneity and productivity [19,20].

The morphological variation of *Lolium multiflorum* L. is interpreted based on its high phenotypic plasticity, the ability of an organism to change its phenotype in response to changes in the environment. This is the ability to alter its physiological characteristics in response to environmental variations and its hybridisation (such as L. *multiflorum* × L. *perenne*). This highlights the need for effective localised control methods for this genus’s different species and hybrids, thanks to its competitive nature, which makes it prone to developing resistance to herbicides. The application of pesticides that are harmful to forage and soil while reducing the occurrence of crown rot under different management conditions has resulted in diverse variations in height, green forage, and dry matter yields, which have a significant impact on soil health and associated microbiota, altering the availability of essential nutrients for plant growth and development, affecting their long-term sustainability [13,21].

It has been demonstrated that annual ryegrass can exhibit enhanced quality when subjected to water stress. Landscape heterogeneity indicates that, in specific locations, adding nutrients and soil amendments can significantly regulate pH and create favourable conditions for increased forage production. This is supported by evidence from studies by Abraha et al. (2015) [22] and Bridges et al. (2019) [23].

The Poaceae, including *Lolium multiflorum* L., play a significant role in the remediation of degraded soils. They mitigate this process’s deleterious effects on the environment. They facilitate the restoration of soils that can be utilised for agricultural production in the context of progressive soil degradation and depletion [24,25]. Cultivated as a cover crop, they have been demonstrated to enhance soil fertility, mitigate erosion, and curtail nutrient loss [26,27]. The sustainable management of soil offers solutions to conservation and the reduction in pollution by utilising plants adapted to the local climate and specific pollutants. These plants demonstrate effective responses to drought and salinity stress at the morphological, physiological, and biochemical levels. Consequently, the Poaceae are indispensable not only for sustainable food production but also for protecting the environment and managing ecosystems [28,29].

The appropriate requirement is fertiliser in ryegrass ‘CE’ ranging between 100–150 kg. ha^−1^ of N, 40–60 kg. ha^−1^ of P and 60–80 kg. ha^−1^ of K. The correct application is crucial to increase yields and improve soil quality by maximising the yield and nutrient absorption of *Lolium multiflorum* L. in acid soils (pH 5.1–6.1).

The production of ‘CE’ ryegrass in Peru has been optimised by adopting sustainable technologies and practices. Soil preparation starts with conventional or minimum tillage, complemented by weed control using selective herbicides or mechanical methods. Fertilisation is based on a soil analysis to determine the appropriate application of nitrogen (N), phosphorus (P), potassium (K), and micronutrients using broadcast or fertigation systems. Sowing is conducted at the beginning of the rainy season (March–April) by direct or broadcast sowing; crop management incorporates sprinkler or drip irrigation, rotation with legumes to improve the soil, and controlled grazing in animal production systems. Harvesting occurs when the ryegrass reaches 20–30 cm in height, using hand or mechanical cutting. For conservation, ryegrass can be stored as hay or silage, maintaining its nutritional value for the dry season. These practical recommendations can be readily implemented, thereby optimising milk production and farmers’ profits while reinforcing the significance of proper soil management for attaining more efficient and sustainable agricultural practices [30,31].

## 2. Results

### 2.1. Yield Performance

Table 1 shows the superiority of the genotype LM-58 over LM-43 in terms of green forage yield (20.97 Mg. ha^−1^), dry matter or biomass (4.49 Mg. ha^−1^), and seed yield (259.23 Kg. ha^−1^). Furthermore, the genotype LM-58 exhibited superior performance in terms of protein value (13.48%), with the majority of locations exceeding 20 Mg. ha^−1^ of green forage and 12% protein, representing a more advantageous option for optimising both agricultural productivity and forage quality.

### 2.2. Cutting Time

Table 2 shows the crucial interactions between cutting days and biomass and protein production, highlighting the significance of our results that by increasing the cutting interval from 30 to 60 days, biomass produced increases significantly from 18.87 to 49.10 Mg. ha^−1^. yr^−1^ (*p* < 0.001); similarly, protein production increases from 2.8 to 4.9 Mg. ha^−1^. yr^−1^ (*p* = 0.0405). These results underline the direct influence of cutting time in optimising biomass and protein production, suggesting that a 60- and 45-day cutting interval may be more beneficial for maximising both parameters in the crops evaluated.

There is an effect of the interaction between cutting time (C) and location (L) on the biomass and protein of ryegrass ‘CE’. Figure 1 shows in detail the interaction between C × L, highlighting its effect on the biomass and protein indicators of ryegrass ‘CE’. Panel (A) shows the biomass production expressed in Mg/ha/year. At the same time, panels (B), (C) and (D) show the protein ratios in percentage (%), Mg/ha/cut and Mg/ha/year, respectively, for the genotype LM-58. This analysis shows that the Baños del Inca site had the best yields in terms of biomass and protein content in Mg/ha/yr, highlighting the importance of cutting time to optimise yield and quality of ryegrass ‘CE’ crops, providing relevant information for farm management and the improvement of cropping practices. In addition, the interaction for protein percentage (*p* = 0.0036) is an important observation to note.

### 2.3. Morphological Characteristics

Table 3 shows the differentiation and morphological characterisation of ryegrass ‘CE’ for the genotypes LM-58 and LM-43. The results highlight the superiority of the LM-58 genotype, which showed better results in the evaluated characters compared to the LM-43 genotype, which obtained lower results in all of these parameters.

Table 4 shows a comprehensive morphological characterisation of the qualitative characteristics of the genotype LM-58 of ryegrass ‘CE’ This analysis reveals the diversity and complexity of the morphological characters of the genotype, which provides a detailed and multifaceted view of its behaviour and physical attributes throughout its growth and development.

Table 5 shows the evaluation of the qualitative characteristics of the genotype ryegrass ‘CE’ LM-58, which was carried out with meticulous attention to detail; the results showed total uniformity in several aspects, where no atypical plant showed a different colour for the “colour of the lemma”, which indicates uniformity in this character. Similarly, we observed uniformity and the absence of atypical seeds for the presence of “kernel ridge”, with no variation in the level of expression. This assessment was carried out visually by observing 30 plants, confirming their homogeneous classification in both assessment seasons.

Table 6 shows that the vegetative cycle of ryegrass ‘CE’ LM-58 exhibits an impressively early expression at 105 days, with coefficients of variability of 2.80% and 2.73% for the first and second seasons, respectively. Similarly, the “time of flower initiation” demonstrated an early expression at 40 days, with coefficients of variability of 2.91% and 2.59% for the first and second seasons, respectively. The classification of the variety as uniform according to UPOV regulations is supported by the observation that both characteristics exhibit coefficients below the permitted limit of 3% for self-pollinated varieties.

A bioinformatic analysis was conducted to identify the mitochondrial DNA sequences of ryegrass ‘CE’, genotype LM-58, utilising advanced tools to evaluate and compare genetic sequences. The analysis revealed that the ITS1 sequence haplogroup (ITS1 = INIA LM-58) differs from the reference sequence of the ryegrass ‘CE’ by three mutational steps. The electropherograms were analysed using the SeqTrace program to obtain sequences with a minimum confidence value, ensuring the sequences were of high quality. Furthermore, consensus sequences were generated and aligned using the Muscle algorithm in Mega 10.1.6 and confirmed against the Genebank database using the BLASTn tool. This was carried out to ensure the identity of the sequences as Lolium multiflorum L. and to confirm the mitochondrial DNA’s Internal Transcribed Spacer (ITS) region. Phylogenetic analysis was employed to group the study samples into a phylogenetic tree using neighbour-joining and maximum likelihood. This analysis demonstrated that all three samples belong to a unique haplotype that differs from the Genebank reference sequence.

Consequently, haplogroup differences are essential for understanding the species’ evolutionary history and ecological adaptation. This analysis is of significant importance to plant genetics and evolution, as it highlights the potential applications of genetic variability in evolutionary and breeding studies. Please refer to the Appendix A for further details.

## 3. Discussion

In this study, we evaluated the morphological differentiation, productive yield, and cutting time of ryegrass ‘CE’ under the challenging acid soil conditions of the northern highlands of Peru. Our impressive results indicate that the Cajamarquino ecotype adapts to unfavourable conditions. This adaptability is reflected in its morphological characteristics and productive yield. Even when faced with the primary source of abiotic stress, drought, which can severely impact plant growth and development, the ryegrass ‘CE’ produced favourable results regarding green forage and dry matter yields.

The findings of this study are of significant value as they provide invaluable insights into the productive performance of ryegrass ‘CE’ under acidic soil conditions in the northern highlands of Peru. The knowledge gained in this study can be used to inform recommendations on genotype management practices and cutting times, as well as the importance of yield differentiation under different climatic conditions.

### 3.1. Performance Yield

Green forage yield is influenced and conditioned by several factors, such as climate, grazing frequency and intensity, and cutting time [32,33]. Ryegrass is grown in winter because of its adaptability to various soils; this cover crop is valued for its cold hardiness, extensive root system, and high biomass production. It is essential to remember that plant productivity is closely linked to the mineralisation of soil nutrients and their uptake and utilisation by plants. This process involves intense interactions between roots, microflora, and soil fauna [26,34].

It is worth noting that *Lolium multiflorum* L. boasts a higher nutritional value than *Lolium perenne*. Its rich concentration of vitamins and antioxidants makes it a highly sought-after species in the market, offering significant health benefits to consumers [35].

The dry matter yield of ryegrass is influenced by several factors, including soil moisture level, type and method of planting, number of harvests, and fertilisation. Also, the soil texture and the competitive ability of *Lolium multiflorum* to grow on different soil types are factors on which the dry matter productivity will depend [36,37].

It is noteworthy that in the results obtained and as stated by the authors, a particular genotype of ryegrass ‘CE’ exhibits superior performance in terms of green fodder yield. Although both genotypes, LM-58 and LM-43, are Cajamarquino ecotypes, the LM-58 genotype exhibits a significantly higher yield compared to the LM-43 genotype. This dominance can be attributed to the superior characteristics exhibited by the LM-58 genotype, including greater leaf length and width, intense and dark leaf colour, and greater plant height after vernalisation, among others. These characteristics, which are corroborated in detail in Table 3 and Table 4, indicate not only a higher yield in terms of biomass produced, but also a higher efficiency in nutrient uptake. In addition, the LM-58 genotype has shown greater resilience to adverse climatic and soil conditions, which is vital for its adaptation and productivity in the locations evaluated. The above characteristics make the LM-58 genotype a more viable and promising option for farmers seeking to maximise green forage production.

As for the locations evaluated, despite being in geolocations with similar conditions, green forage yields varied from location to location; this could be due to differences in cropping seasons across years and evaluation periods. Factors such as microclimatic variations, specific agronomic practices, and soil quality may have influenced these results; however, the Baños del Inca location showed the highest yield in both production seasons, as detailed in Table 1, Table 6 and Table 8. This consistency suggests that Baños del Inca offers more favourable conditions for growing ryegrass ‘CE’ than the other locations. These findings highlight the importance of carefully selecting locations to maximise crop production and suggest that Baños del Inca could serve as a model for successful agricultural practices in the region.

### 3.2. Cutting Time

The analyses revealed that the forage quality of the ryegrass ‘CE’ is determined by the moment of cutting according to the dry matter (DM) yield per hectare and the percentage of protein. Three groups of analysis were formed with samples every 30, 45, and 60 days, showing that at 30 days, the best result was obtained with a protein content of 14.84%, as opposed to 45 days; while biomass increases in this period, protein content decreases to 12.74%, indicating that the optimal cutting time for maximizing forage quality is at 60 days to achieve a more efficient productive use, given that there are no differences in annual protein production at the mowing ages of 60 and 45 days.

Studies have shown that 90% of ryegrass seedlings emerging on the first day compared to other species is due to the ability of the mowing period to manage ryegrass dominance. The effect of the timing of first and repeat mowing on seedling growth depends on the impact of mowing height, as defoliating based on soil height minimises damage to the growing points of the tillers, with 22–50% of the variability in growth at 60 days observed to be due to these initial treatments [38,39].

This study shows that cutting times are critical factors in determining the quality and quantity of ryegrass ‘CE’ forage. A 30-day cutting interval is optimal for maximising protein content, while an adequate cutting height can significantly improve grass recovery and growth. Furthermore, the variability in green forage yield between sites underlines the importance of taking local conditions into account in agronomic practices. Finally, it should be noted that the genotype LM-58, in interaction with the locations evaluated, obtained a significant value (*p* = 0.0036) for protein percentage. These data provide valuable information for genetic improvement and selection in forage production and show that this genotype outperformed the other local genotypes, as confirmed in Table 2.

### 3.3. Morphological Differentiation

Morphological differentiation among local genotypes of ryegrass ‘CE’ is essential to understanding its adaptation and performance under various environmental conditions. The study results indicate that the genotype LM-58 has superior characteristics to the genotype LM-43. In the parameters evaluated for growth habit, LM-58 exhibited semi-erect growth and a greater intensity of dark green colour in its leaves, contrasting with the more upright growth and light green colour of LM-43. In addition, LM-58 showed a greater width of the last leaf (0.8 cm vs. 0.6 cm for LM-43) and a higher number of seeds per spikelet (13 vs. 10 by LM-43). The growth rate of LM-58 was also higher, reaching 1.6 cm per day compared to 1.2 cm per day for LM-43, allowing faster recovery after cutting and higher forage production in a shorter period. Age at the fourth leaf was earlier in LM-58 (28 days) than LM-43 (35 days), indicating the greater earliness of LM-58, which may benefit faster production cycles.

Bioinformatic analysis of mitochondrial DNA sequences shows the identity of ryegrass ‘CE’ and identified two distinct haplotypes. The LM-58 genotype clustered into one haplotype together with other samples under study, differing from the reference sequence of the LM-43 genotype. Cross-pollinated species, such as ryegrass, show high genetic variability within and between populations. Thus, *L. multiflorum* and the wind dispersal mating system promote high genetic variability within populations but low genetic differentiation between populations [40,41]. This genetic variability explains why the haplotypic network constructed shows two unique haplotypes, haplotype 1 containing the three samples under study and haplotype 2 containing the reference sequence, thus reinforcing the genetic differentiation of the LM-58 genotype, as can be seen in the Appendix A. Finally, the LM-58 genotype showed increased resilience to adverse climatic and soil conditions, vital for its adaptation and productivity in the locations evaluated. These characteristics make LM-58 a more viable and promising option for maximising green forage production.

### 3.4. Fodder Preparation, Handling and Storage

#### 3.4.1. Soil Preparation and Sowing

Soil preparation is crucial for the successful establishment of ryegrass ‘CE’; tillage is recommended to allow good soil drainage and aeration. This may include ploughing and harrowing to create a fine seedbed. Sowing should be carried out at the right time, usually at the beginning of the rainy season, to ensure effective germination. Direct seeding or traditional techniques such as broadcasting can be chosen, depending on the size of the area and soil conditions. Proper soil preparation contributes to faster and more vigorous establishment and development of ryegrass ‘CE’.

#### 3.4.2. Crop Management

Crop management of ‘CE’ ryegrass requires adequate fertilisation, focusing mainly on N, P, and K supply, which is essential for rapid growth and development of high-quality forage. Irrigation is another critical factor, especially in regions with insufficient rainfall, where sprinkler or drip systems are used to maintain soil moisture. In addition, it is important to implement effective pest and disease control through crop rotation, using resistant varieties and applying pesticides if necessary. Controlled grazing is part of good crop management, where it ensures that forage is cut and consumed at a suitable height to promote regrowth and thus maximise the use of the pasture without overgrazing the land.

#### 3.4.3. Forage Storage

Adequate storage of ryegrass ‘CE’ is essential to maintain its nutritive value when fresh grass is unavailable. Forage can be harvested and stored as hay or silage. For hay, it is essential to cut the ryegrass at the optimum time of growth, usually when it reaches 20 to 30 cm in height, and to dry it quickly to avoid loss of nutrients. Silage, on the other hand, requires the grass to be chopped and compacted in silos to ferment under anaerobic conditions, which preserves the quality of the forage and improves its digestibility. These storage methods allow producers to have a constant supply of high-quality forage throughout the year, thus improving the sustainability of livestock production.

## 4. Materials and Methods

### 4.1. Place and Duration of Study

This research was conducted in the Cajamarca Region, Peru, and had two experimental stages (Table 7). The first phase was developed in the agricultural campaigns of 2017–2018; the second campaign was created in the period 2018–2019, where the productive yield parameters (green forage, dry matter, and time of cutting) were evaluated in two local genotypes, LM-43 and LM-58, as detailed in Table 8, and was developed in 6 locations located in the region of Cajamarca, department of Cajamarca, Peru, with an average annual temperature of 15.6 °C, average relative humidity of 79%, and an average annual rainfall of 2963 mm. yr^−1^ [42]. This research was evaluated in December, a suitable time for installing this forage species.

### 4.2. Experimental Units

To determine the productive yield of ryegrass ‘CE’ a Completely Randomised Block Design (CRBD) was employed at 6 locations within the Cajamarca region. This design, renowned for its robustness and reliability, proved to be the optimal choice for our experiment. The ryegrass ‘CE’ LM-58 and LM-43 genotypes were evaluated; each plot was 6 m × 5 m, and each genotype was cultivated in 2 experimental plots at each location.

The treatments were divided according to the phenological stage of the ryegrass ‘CE’, where T1 = genotype LM-58 and T2 = genotype LM-43. It is essential to mention that the sowing system was implemented in the 6 locations with a sowing density of 30 kg. ha of seed. Fertilisation was carried out in two seasons, each with two key fertilisation moments: in the first fertilisation at sowing, agricultural Urea (N) = 80 kg. ha^−1^. yr^−1^, Triple Calcium Superphosphate (P_2_O_5_) = 130.4 kg. ha^−1^. yr^−1^, and Potassium Chloride KCl (K_2_O) = 66.4 kg. ha^−1^. yr^−1^, and in the second tillering fertilisation, which consisted only of Urea = 80 kg. ha^−1^. yr^−1^, using the same fertilisation rates for both campaigns.

### 4.3. Experimental Material

The experimental material was derived from a backcrossing process involving a population of previously selected elite genotypes. The experimental treatments comprised the genotypes LM-58 and LM-43 of ryegrass ‘CE’, which underwent two selection processes. The initial phase involved 680 accessions, while the subsequent phase encompassed 21 exceptional emerging lines selected over 25 years.

Table 8 presents the detailed periods and critical activities of the study, such as the initial selection based on growth habit, anthesis, and pest and disease resistance, followed by propagation and polycrossing to enhance desirable traits. The research, which began in 1995 with the collection and germination of seedlings, progressed through several phases of characterisation, selection, and field trials. After numerous tests, the genotype LM-58 was identified as an elite genotype of ryegrass ‘CE’, marking a significant milestone in our research.

### 4.4. Sampling

We evaluated the morphological differentiation, yield, and cutting time of the ryegrass ‘CE’ genotypes LM-58 and LM-43 with utmost precision. We obtained the samples using a square metre, manually cut the forage with a sickle, and weighed it fresh on an electronic balance. A 100 g sample was then taken to the laboratory for further analysis, including determining the percentage of dry matter and calculating biomass yield.

The number of seeds harvested was recorded per square metre, while plant height was measured with a ruler, averaging the values for each one after forage cutting. Six samples were taken for each treatment to determine the best cutting time according to the phenological stage of the associated plant. These samples were left to wilt for 12 h, reduced to 5 cm in size, and then placed in 50 kg polyethene bags.

For the chemical analysis, we followed the rigorous AOAC methodologies (AOAC, 1990) to ensure the accuracy and reliability of our results. The samples were dried in a forced-air oven, ground to a specific particle size, and then analysed for crude protein (CP), Crude Fibre (CF), Ethereal Extract (EE), ash, and Free Nitrogen Extract (FN).

For seed production, a uniform cut was made before flowering to achieve uniform and synchronised growth between flowering and fruiting.

### 4.5. Evaluation Characteristics

Following the comprehensive guidelines of the Union for the Protection of New Varieties of Plants (UPOV), the evaluation process was conducted at a single location and over two similar vegetation periods. Under these stringent conditions, the local genotypes LM-58 and LM-43 underwent a thorough evaluation in two cycles of identification trials.
-Cycle 1: In the 2017–2018 season.-Cycle 2: During the 2018–2019 campaign.

According to the regulations for generating new cultivars established in Decision 345 in Article 10° and as established by the Head Resolution N° 047-2000-INIA, Chapter VII, Article 12, a variety is considered distinct if one or several distinctive characteristics are notably different from any other commonly known variety. It is sufficient that only one characteristic is distinct for the variety to be considered distinct, Article 21. The distinctness test, a crucial part of the evaluation process, must include the commonly known varieties with a more remarkable similarity to the variety for which protection is sought. For quantitative characteristics, the hypothesis of equality will be evaluated, and only if this hypothesis is rejected with a significance level of 5% probability will it be possible to conclude that the varieties are different. If the characteristic is qualitative, the variety to be protected must show a different characteristic status compared to other varieties.

The evaluations of the genotypes LM-58 and LM-43 were carried out according to the UPOV Codes: LOLIU_PER; LOLIU_MUL_ITA; LOLIU_MUL_WES; LOLIU_BOU; and LOLIU_RIG. These codes, internationally recognised standards for the distinctness, uniformity, and stability (DUS) testing of plant varieties, were applied to the parameters for cultivating semi-perennial and perennial forage grasses. The criteria established by the International Union for the Protection of New Varieties of Plants [43] were used as a reference, and the focus was on the morphological characterisation of ryegrass ‘CE’, local cultivars LM-58 and LM-43. The evaluations unveiled these grasses’ fast-growing and robust nature, signalling their promising adaptability to various soil types and climatic conditions. They also demonstrated efficient nutrient uptake and soil anchorage, further bolstering their potential.

This bioinformatic analysis of ryegrass ‘CE’ DNA involves the extraction, sequencing, and analysis of the genetic material of this plant to understand its genomic structure, characteristics, and genetic variability. The extraction was performed with specific kits that allow high-quality genomic DNA to be obtained from pulverised plant tissues and processed with lysis buffers containing chaotropic salts, denaturing agents, and detergents. The extracted DNA is purified by filtration and centrifugation to remove polysaccharides, contaminants, and cellular debris, using binding and washing buffers to ensure the removal of proteins, RNA, metabolites, and other PCR (Polymerase Chain Reaction) inhibitors. The purified DNA is used in subsequent reactions, such as PCR, which allows the amplification and detailed analysis of specific gene sequences to study the structure and function of the ryegrass genome. Once sequences are obtained, bioinformatics analysis includes sequence alignment, gene identification, functional annotation, and comparison with existing genomic databases, helping to identify genetic variations, evolutionary relationships, and potential applications in breeding and biotechnology. This knowledge has inspiring applications in crop improvement, such as developing varieties with increased disease resistance, tolerance to adverse climatic conditions, and improved forage nutritional quality, thereby contributing significantly to global food security. This complex analysis combines high-quality DNA extraction and purification with advanced sequencing techniques and computational analysis to reveal crucial information about the biology and potential of this important plant species.

Table 9 presents the general parameters for the assessment of the qualitative morphological characteristics applied to the genotypes LM-58 and LM-43 of ryegrass ‘CE’, covering various physical and developmental attributes of the genotype’s evaluation. This characterisation will highlight both differences and similarities between the genotypes, providing valuable information for the implementation of breeding programmes and the optimised agronomic practices detailed in Table 3.

### 4.6. Statistical Analysis

An initial exploratory analysis of the data was conducted to ascertain the normality and homogeneity of variances. This was achieved through the utilisation of the Shapiro–Wilks (*p* < 0.05) and Levene (*p* < 0.05) tests, respectively. To ascertain the differences between the genotypes (LM-58 and LM-43), the cutting frequency (30, 45, and 60 days), the six locations, and their interaction in the different yield parameters and nutritional aspects, an Analysis of Variance (ANOVA) was performed, using Infostat software (version 2020). To determine the significant comparison of means, the Duncan test (*p* < 0.05) was employed.

## 5. Conclusions

The results obtained in this research have fundamental implications for agriculture in regions with acid soils due to the extraordinary capacity of ryegrass ‘CE’ to maintain an excellent productive yield under adverse conditions, which suggests that it is a viable option to improve forage production in the northern highlands of Peru. It will contribute to the sustainability of diverse agricultural practices and improve rural communities’ sustainability, resilience, and economies. Correctly managing cutting time and fertilisation is critical to maximise forage yield and quality. The study of ryegrass ‘CE’, under acid soil conditions in the northern highlands of Peru has yielded essential results with important implications for agriculture in these regions. The ryegrass ‘CE’ LM-58 showed remarkable adaptability to the different geographical conditions of the six sites studied, maintaining high green fodder and dry matter yields and suggesting its viability for improving fodder production in these areas. The detailed analysis of the morphological characteristics of the LM-58 and LM-43 genotypes revealed valuable differences and similarities for the implementation of breeding programs and optimised agronomic practices, highlighting the superiority of the LM-58 genotype for its greater resilience and efficiency in the production of significant biomass yields, with a protein content of 14.84% at 30 days and 12.75% at 45 days. In addition, bioinformatic analysis of mitochondrial DNA sequences identified two distinct haplotypes, reinforcing the genetic differentiation of LM-58 and consolidating it as a promising option for farmers seeking to maximise green fodder production. In conclusion, the LM-58 genotype of the ryegrass ‘CE’ is a species well adapted to local livestock production, capable of maintaining excellent productive performance, and represents a viable alternative for improving forage production in the highlands of northern Peru.

## Figures and Tables

**Figure 1 plants-13-02331-f001:**
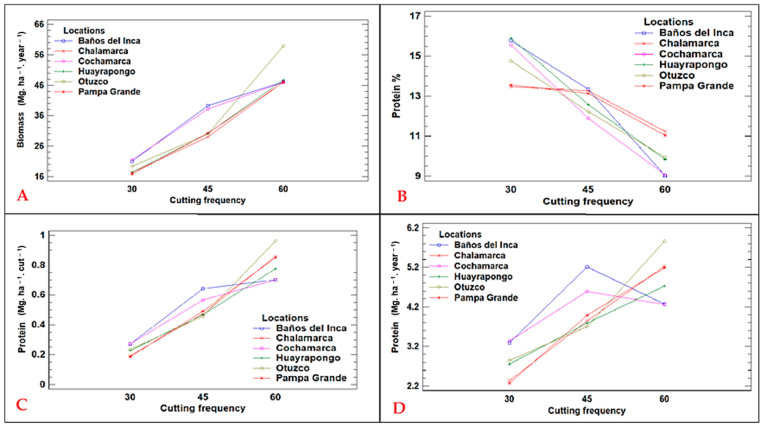
Interaction between cutting time and (**A**) annual biomass (Mg. ha^−1^. yr^−1^), (**B**) protein percentage (%), (**C**) protein yield per cut (Mg. ha^−1^. yr^−1^), and (**D**) annual protein yield (Mg. ha^−1^. yr^−1^) for ryegrass ‘CE’ genotype LM-58.

**Table 1 plants-13-02331-t001:** Yield performance of the two ryegrass ‘CE’ genotypes at the six locations.

Factors	Green Forage (Mg. ha^−1^)	Biomass (Mg. ha^−1^)	Seeds (kg. ha^−1^)	Protein (%)
Genotype				
LM-43	20.15 b	3.64 b	189.95 b	11.95 b
LM-58	20.97 a	4.49 a	259.23 a	13.48 a
SE	0.21	0.06	13.56	0.29
Locations				
Baños del Inca	22.19 a	4.3 a	251.98	13.31
Chalamarca	18.83 c	3.95 b	218.24	12.50
Cochamarca	21.26 ab	3.77 b	207.95	13.04
Huayrapongo	22.10 a	4.35 a	217.40	12.18
Otuzco	20.71 b	4.06 ab	254.75	13.34
Pampa Grande	18.26 c	3.95 b	197.20	11.94
SE	0.36	0.11	23.49	0.48
*p*-value				
Genotype (G)	0.0104	<0.001	0.0047	0.0037
Locations (L)	<0.001	0.0061	0.8743	0.7580

SE: Standard error. Different letters in each factor and in each column represent significant differences (Duncan test, *p* < 0.05).

**Table 2 plants-13-02331-t002:** Biomass and protein by cutting times for genotype LM-58.

Factors	Biomass		Protein		
	Mg. ha^−1^. cut^−1^	Mg. ha^−1^. yr^−1^	%	Mg. ha^−1^. cut^−1^	Mg. ha^−1^. yr^−1^
Cutting time					
30 days	1.55 c	18.87 c	14.84 a	0.23 c	2.80 b
45 days	4.05 b	32.83 b	12.74 b	0.52 b	4.19 a
60 days	8.07 a	49.10 a	10.03 c	0.81 a	4.91 a
*SE*	0.24	*1.60*	0.44	*0.036*	*0.26*
*p*-value					
Cuts (C)	0.0000	0.0000	0.0001	0.000	0.0006
Locations (L)	0.4008	0.5376	0.9755	0.9599	0.8896
C × L	0.6303	0.6192	0.0036	0.2076	0.1817

Different letters in each factor and in each column represent significant differences (Duncan test, *p* < 0.05).

**Table 3 plants-13-02331-t003:** Morphological differentiation of ryegrass ‘CE’ LM-58 and LM-43.

Characters	LM-58	LM-43
Level of Expression	Forms of Growth	Level of Expression	Forms of Growth
Growth habit	Intermediate	Semi-erect	Medium	Erect
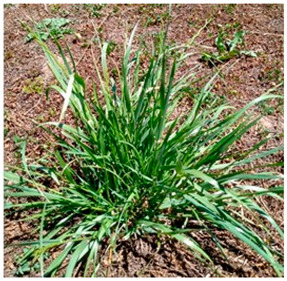	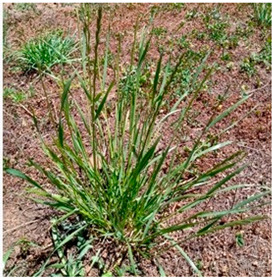
Intensity of green colour (leaf)	Dark	Dark green	Medium	Light green
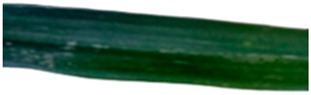	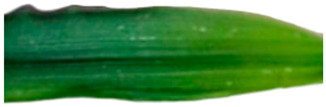
Width (last sheet)	Wide (0.6 to 0.8)	0.8 cm	Medium (0.4 to 0.6)	0.6 cm
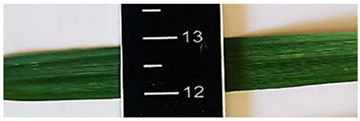	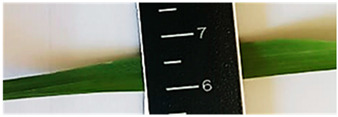
Number of seeds per spikelet	Very high > 11	13	High (9 to 10)	10
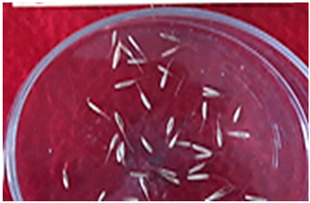	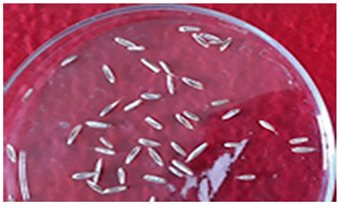
Growth rate per day (cm)	Early	1.6	Intermediate	1.2
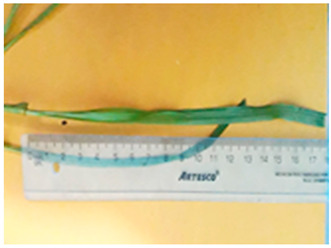	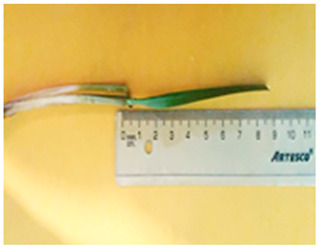
Age at fourth leaf (days)	Early (30 to 35)	28	Intermediate (35 to 40)	35
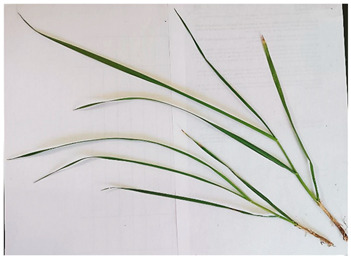	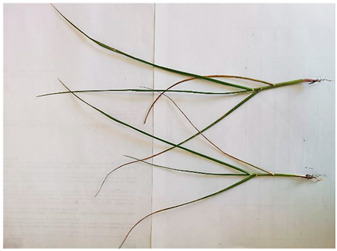
Basal area in cm (basal coverage)	Medium (70 to 100 cm^2^)	78.5 cm^2^	Medium (70 to 100 cm^2^)	66.6 cm^2^
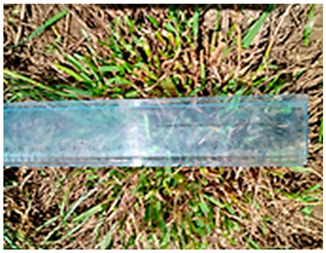	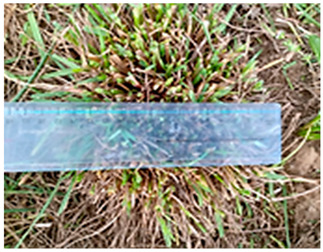
Length-to-width ratio (last leaf)	Medium	Medium (32 L. × 0.8 A.)	Medium	Medium (28 L. × 0.6 A.)
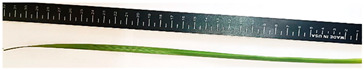	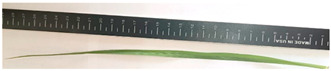
Inflorescence, spikelet density	Medium (35)	±0.7	Medium (28)	±0.9
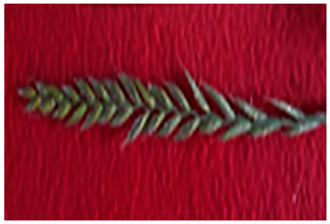	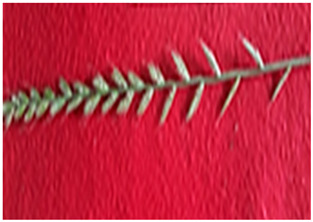
Expression of the ligule	Notorious	Visible	Notorious	Visible
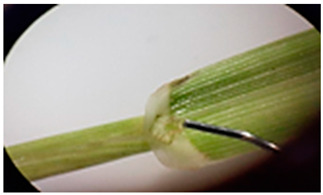	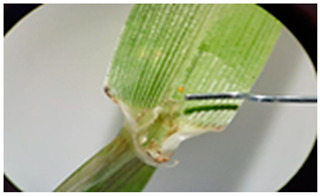
Expression of atria	Notorious	Visible	Notorious	Visible
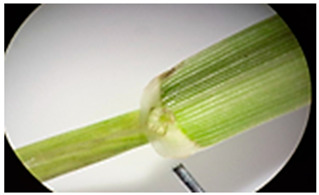	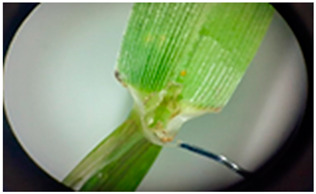

**Table 4 plants-13-02331-t004:** Qualitative morphological characterisation of the assessed qualitative characteristics LM—58.

Character	Observation Period	States	Score
Growth habit	Plant	Semi-erect	3
Leaf length in vegetative state (33 to 38)	Leaflet	Long	7
Leaf width in vegetative state (0.8–1.0)	Limbo	Media	5
The intensity of the green colour	Leaflet	Dark	7
Width after vernalisation (basal coverage)	Plant	Medium	5
Height (after vernalisation)	Plant	High (60–70)	7
Tendency to form inflorescences (without vernalisation)	Plant	Medium	5
Flag leaf length	Leaflet	Medium	5
Width	Leaflet	Wide (0.6 to 0.8)	7
Length/width ratio	Leaflet	Medium	5
Length of most extended stem, including inflorescences (when fully developed)	Stem	Long (45 to 55)	7
Length including inflorescences (when fully developed)	Panicle	Long (25 to 30)	7
Number of spikelets	Panicle	Very high (>11)	9
Spikelet density	Panicle	Medium	5
External length of basal spikelet	Glumes	Long (0.8 to 1.0)	7
Length of basal spikelet, excluding the ridge	Glumes	Long (1.5 to 2.0)	7
Growth rate (cm/day)	Plant	High (1.4 to 1.8)	3
Form	Leaflet	Lanceolate medium	3
Length at 30 days	Plant	Long (33 to 39)	7
Presence of atria	Leaflet	Medium	5
Expression of the ligule	Limbo	Regularly noted	7
Age at fourth leaf (cm)	Leaflet	High (35 to 40)	5
Number of seeds per spikelet	Grain	High (12 to 14)	7
Short slogan	Grain	Dark cream	5
Length of the locker	Grain	Cut	
Time of beginning of flowering (green fodder)	Flowering	High (60 to 70 days)	5
Growing season (grain production)	Harvest	High (100 to 110 days)	7
Presence of edges	Grain	Present	3

**Table 5 plants-13-02331-t005:** Evaluation of homogeneity, lemma colour, and grain ridge presence of ryegrass ‘CE’ LM-58.

Campaign	Characters					
	Colour of the Slogan			Grain Presence of Edge		
	Level of Expression	Off-Type Plants	Maximum Number	Level of Expression	Off-Type Plants	Maximum Number
First	Dark cream	0	3	Present	0	3
Second	Dark cream	0	3	Present	0	3

**Table 6 plants-13-02331-t006:** Evaluation of uniformity, vegetative cycle, and time of beginning of flowering (days).

Campaign	Characters							
	Vegetative Cycle				Start of Flowering			
	Level of Expression	Average (Days)	SD.	CV.	Level of Expression	Average (Days)	SD.	CV.
First	Semi-early	105.0	2.94	2.80%	Earliness	40.2	1.17	2.91%
Second	Semi-early	105.8	2.87	2.73%	Earliness	39.7	1.03	2.59%

SD: Standard deviation, CV: coefficient of variation.

**Table 7 plants-13-02331-t007:** Geographical location and soil chemical composition of the locations for the yield evaluation of the genotypes LM-58 and LM-43 of ryegrass ‘CE’.

Locations	Agricultural Campaign	Altitude	District	Geographical Coordinates		Chemical Composition of the Soil				
						P (ppm)	K (ppm)	pH	OM (%)	Al (mEq/100 g)
Otuzco	2018–2019	2680	Baños del Inca	−7.131402	−78.469430	16.22	225	4.5	1.54	0.10
Cochamarca	2016–2017	2820	Gregorio Pita	−7.181967	−78.467969	8.11	195	4.3	2.13	0.90
Huayrapongo	2015–2016	2650	Baños del Inca	−7.181967	−78.467970	33.70	271	6.1	3.82	0.00
Pampa Grande	2014–2015	2680	Cajabamba	−7.611059	−78.070150	6.20	170	3.8	3.55	0.50
Chalamarca	2014–2015	2760	Chota	−7.279996	−78.218141	18.22	186	4.9	4.12	0.70
Baños del Inca	2014–2015	2650	Cajamarca	−7.159426	−78.461260	10.90	330	7.0	2.25	0.00

P = phosphorus; K = potassium; OM = organic matter; Al = aluminium; pH = hydrogen ion potential.

**Table 8 plants-13-02331-t008:** Timeline of key activities in the development of elite ryegrass ‘CE’ genotypes.

Period (Years)	Activity	Description
1995	Collection and installation of experimental material.	Sowing (germination beds).
1996–1998	Characterisation of genotypes	Selection and sowing of 680 seedlings (genotypes).
1999–2000	Selection multiplication	Characterisation of genotypes (growth habit, anthesis, and pests and diseases).
2001–2003	Multiplication selection	Characterisation and selection of the best genotypes.
2004–2006	Research	Selection and evaluation of elite genotypes in green forage, dry matter, and seed.
2006–2008	Research	Polycrosses between the 10 elite genotypes in experimental field. To fix superior characters.
2009–2011	Field testing by growers	Selection and evaluation of 10 superior genotypes in forage production, nutritive value, and tolerance to pests and diseases.
2011–2016	Adaptation and efficiency trials in growers’ fields.	First campaign–Producers’ field adaptation and efficiency trials.
2016–2019	Adaptation and efficiency trials in growers’ fields.	Second campaign—Adaptation and efficiency trials in growers’ fields.

**Table 9 plants-13-02331-t009:** Qualitative morphological evaluation characteristics of ryegrass ‘CE’ genotypes LM-58 and LM-43.

Parts	Features
Plant	Vegetative growth habit (without vernalisation). Tendency to form inflorescences (without vernalisation). Width after vernalisation (basal area in cm). Height after vernalisation. Length of most extended stem, including inflorescences (when fully developed).
Leaflet	Leaf length (in a vegetative state). Leaf width (in a vegetative state). The intensity of the green colour. Flag leaf length (last leaf). Width (previous sheet). Length/width ratio (previous leaf). Leaf shape. Presence of atria. Expression of the ligule; Motto colour.
Inflorescence	Length, including inflorescences (when fully developed). Number of spikelets. Spikelet density. Length of outer glume of basal spikelet. Length of basal spikelet, excluding the ridge. Number of seeds per spikelet. Grain presence of edge. Time of beginning of flowering.
Other	Growth rate per day. Plant length at 30 days. Growing season for grain seed.

## Data Availability

Data sharing is not applicable to the article as datasets were generated or analysed during the current study.

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
