# Peer review of "Morphological Differentiation, Yield, and Cutting Time of Lolium multiflorum L. under Acid Soil Conditions in Highlands"

_plants, 2024, doi:10.3390/plants13162331_

Round 1

Reviewer 1 Report

Comments and Suggestions for Authors

General comments

I propose to completely rework the Abstract for two reasons:

1) It contains a lot of information that I did not find in Results section.

2) The whole article is focused on the comparison of two genotypes LM-58 and LM-43. There is no mention of these genotypes and the differences between them in the Abstract.

Please correct carefully the units throughout the manuscript - they are usually given incorrectly - for example, in L.125 it cannot be kg.ha, but kg/ha (as is correct in L.126) or kg.ha-1. I will mention the lines 25, 30, 93, 141, Tab.1, 154, Tab.2, 165, 167, Fig.1, 176 etc.

Minor comments

l.20-21 „six genotypes“? I think you were only comparing LM-58 and LM-43.

L.24-25 Is this information included somewhere in the Results section? Where?

L.26-27 I didn't find this information in the Results section.

L.28  Is this information included somewhere in the Results section? Where?

L.30 „4.05 Mg.ha.cut“? I see 0.52 in the Table 2.

L.94-95 Why the same information is duplicated in L. 106-108?

L.142-145 duplicated sentence

„higher production capacity in most locations“ can not be seen from the Table 1

L.173 The charts are for LM-58 and LM-43 combined (like the whole chapter 2.2.)?

L.205 ???

L.215 Explain the abbreviations DS and CV

L.217-221 Please provide more information about the bioinformatic analysis. It's incomprehensible this way.

L.278-279 Explain the difference between this conclusion and conclusion in L. 156-157

L.280-281 Where is this shown in the results?

L.291 „in interaction with the sites“ - What do you mean? I assume that p=0.0036 is the value of the difference between LM-58 and LM-43 at all locations combined.

L.310-312 Please provide a link to the figure in the supplementary material.

L.334 Table 7 is not cited anywhere.

L.344 How many plots of each variety did you have in each location? 2, 6 or 12?

L.423 I do not understand the meaning of Table 9. I would expect differences between LM-58 and LM-43 genotypes.

L.427-426 Levene is test for checking of  homogeneity of  variances and K-S is test for data normality.

L.429 The Duncan test seems inappropriate for your design (2 cycles, 6 locations). More appropriate would be, for example, a multi-way ANOVA or nested ANOVA (with cycle and/or location as a nested factor). In addition, in Table 2, you have petrily used a two-way ANOVA.

Author Response

General comments

Comments 1: [I propose to completely rework the Abstract for two reasons:]

1) It contains a lot of information that I did not find in Results section.

2) The whole article is focused on the comparison of two genotypes LM-58 and LM-43. There is no mention of these genotypes and the differences between them in the Abstract.

Response 1: [The abstract has been subjected to a comprehensive revision and rewriting process, in accordance with the comments and recommendations set forth in the aforementioned commentary].

Comments 2: [Please correct carefully the units throughout the manuscript - they are usually given incorrectly - for example, in L.125 it cannot be kg.ha, but kg/ha (as is correct in L.126) or kg.ha-1. I will mention the lines 25, 30, 93, 141, Tab.1, 154, Tab.2, 165, 167, Fig.1, 176 etc].

Response 2: [The recommendations set forth in the aforementioned commentary have been duly considered and appropriate corrections have been implemented].

Minor comments

Comments 3: 

  • 20-21 „six genotypes“? I think you were only comparing LM-58 and LM-43.
  • 24-25 Is this information included somewhere in the Results section? Where?
  • 26-27 I didn't find this information in the Results section.
  • 28  Is this information included somewhere in the Results section? Where?
  • 30 „4.05 Mg.ha.cut“? I see 0.52 in the Table 2.

Response 3: [Based on the above commentary, the observations presented in the summary have been reviewed and revised according to the predicted recommendations].

Comments 4:

  • 94-95 Why the same information is duplicated in L. 106-108?
  • 142-145 duplicated sentence

Response 4: The repetition of sentences in the aforementioned lines has been removed.

Comments 5: [higher production capacity in most locations “cannot be seen from the Table 1].

Response 5: [The production capacity of the locations has been evaluated as main effects, and show that there are significant differences for green forage (Mg. ha-1) and biomass (Mg. ha-1) between locations but they are not highly variable.]

Comments 6: [L.173 The charts are for LM-58 and LM-43 combined (like the whole chapter 2.2.)?].

Response 6: [This line exclusively pertains to LM-58 at three cut-offs, as it represents the elite genotype in the initial characterization].

Comments 7: [L.205 ???].

Response 7: [Deleted line]

Comments 8: [L.215 Explain the abbreviations DS and CV].

Response 8: [As previously stated in the commentary, the aforementioned abbreviations were duly elucidated].

Comments 9: [L.217-221 Please provide more information about the bioinformatic analysis. It's incomprehensible this way].

Response 9: [As recommended in the suggested comment, further information on bioinformatics analysis has been provided].

Comments 10: [L.278-279 Explain the difference between this conclusion and conclusion in L. 156-157].

Response 10: [The text has been analysed in line with the conclusions set out in the commentary. Both conclusions have been corrected by re-emphasising the importance of cutting time for a more optimal utilisation of protein and biomass].

Comments 11: [L.280-281 Where is this shown in the results?].

Response 11: [The text referenced in the aforementioned commentary pertains to the cited literature and has been modified to prevent any potential confusion].

Comments 12: [L.291 „in interaction with the sites“ - What do you mean? I assume that p=0.0036 is the value of the difference between LM-58 and LM-43 at all locations combined].

Response 12: [This line pertains to the interaction value between the locations evaluated with the specified cut-off time]. 

Comments 13: [L.310-312 Please provide a link to the figure in the supplementary material.]

Response 13: [The external link is not considered, because it would be linked as a genuine file to the article at the time of publication.]

Comments 14: [L.334 Table 7 is not cited anywhere].

Response 14: [Table 7 has been referenced in accordance with the commentary observation].

Comments 15: [L.344 How many plots of each variety did you have in each location? 2, 6 or 12?].

Response 15: [A total of 12 plots were assessed, two plots at each site.]

Comments 16: [L.423 I do not understand the meaning of Table 9. I would expect differences between LM-58 and LM-43 genotypes].

Response 16: [Table 9 provides a detailed overview of the qualitative characteristics to be evaluated for both ryegrass 'CE' genotypes. For a more comprehensive understanding of these characteristics, please refer to Table 3].

Comments 17: [L.427-426 Levene is test for checking of homogeneity of variances and K-S is test for data normality. L.429 The Duncan test seems inappropriate for your design (2 cycles, 6 locations). More appropriate would be, for example, a multi-way ANOVA or nested ANOVA (with cycle and/or location as a nested factor). In addition, in Table 2, you have petrily used a two-way ANOVA].

Response 17: [The statistical procedures performed in the data analysis - using ANOVA - have been described.]

Reviewer 2 Report

Comments and Suggestions for Authors

The present study afford very interesting and useful information for Italy ryegrass production and utilization in Peru. I enjoyed reading the ms, what I concerned is the total biomass/yield/benefical in a whole growth season, particular the cutting times related biomass/yield. E.g., the cutting frequency seems reduced almost half of normal cuttings. The author mostly compared the results between single cuttings.

Introduction: The logic need improved, there are too much of statement of ryegrass in several parts. The logic could be e.g, the problems in Peru grassland and livestock, why ryegrass, the gap, the aims of present study, with hypothesis. Focus on the parameters present study involved.

Delete the information that is not related with the parameters and the content of present study, e.g, the grazing, the weed, the germination, the water stress. What is the relationship of these with your study?  

line 132-137, The significance of present study, move to discussion.

Line 58: “and temperatures between 12 and 18 °C”, please check the range of temperature. I assume the range wider than 12-18.

Line 60: change Varieties to varieties.

Line 149: More details for a, b .

Line 153-156: The cutting time changed from 30 to 60, which means the cutting frequency reduced half, how about the total biomass of the growth season? Line 156, suggesting that a 60-day cutting interval may be more beneficial for maximising both parameters in the crops evaluated, should be very careful.

Line 216-222, more details of the mitochondrial results.

I like Table 7. Very good summary and afford very useful information for the design of present study.

Did the author afford the methods of mitochondrial? I failed find the content.

Author Response

Comments 1: [This study provides very interesting and useful information for the production and utilisation of Italian ryegrass in Peru. I enjoyed reading the manuscript. What interests me is the total biomass/yield/profit over a whole growing season, in particular the biomass/yield related to cutting times. For example, mowing frequency seems to be reduced by almost half of normal mowing. The author mainly compared the results between individual cuts].

Response 1: [The cutting samples were evaluated individually in order to ensure greater precision in the study, based on cutting time. This was done in order to ascertain which cutting exhibited the optimal performance in accordance with the evaluated parameters, including biomass and protein content].

Comments 2: [Introduction: The logic needs to be improved, as there are too many statements about ryegrass in several parts. The logic could be, for example, the problems of rangeland and livestock farming in Peru, why ryegrass, the gap, the objectives of the present study and the hypothesis. Focus on the parameters involved in the present study.

Delete information that is not related to the parameters and content of the present study, e.g. grazing, weeds, germination, water stress. How do these relate to your study?]

Response 2: [In consideration of the comments that have been provided, it has been proposed that the manuscript would be enhanced by a more structured and coherent presentation].

Comments 3: [Line 132-137, The importance of the present study, move to discussion].

Response 3: [In accordance with the proposed commentary, the section from line 132 to line 137 has been relocated to the discussion section].

Comments 4: [Line 58: ‘and temperatures between 12 and 18 °C’, please check the temperature range. I assume the range is wider than 12-18].

Response 4: [The temperature range for 'EC' ryegrass growth has been modified to encompass a wider average range from cold to warm temperatures, as evidenced by the data.]

Comments 5: [Line 60: change Varieties to varieties].

Responses 5: [The term proposed in the aforementioned commentary has been modified].

Comments 6: [Line 149: More details for a, b].

Responses 6: [The comment in question has been duly considered and amended in accordance with the relevant criteria].

Comments 7: [Line 153-156: Cutting time changed from 30 to 60, meaning that the cutting frequency was halved, what about the total biomass of the growing season? Line 156, suggesting that a 60-day cutting interval may be more beneficial to maximise both parameters in the crops assessed, needs to be very careful].

Responses 7: [As indicated in the aforementioned commentary, the lines in question have undergone a re-evaluation and the context has been enhanced for enhanced comprehension. Furthermore, due diligence has been exercised in the delineation of the cutting time and the evaluated parameters].

Comments 8: [Line 216-222, more details of the mitochondrial results].

Responses 8: [In accordance with the aforementioned commentary, the mitochondrial results have been elucidated in greater detail in the preceding lines].

Comments 9: [I like Table 7. Very good summary and provides very useful information for the design of this study].

Responses 9: [We would like to express our gratitude for your favourable comments regarding our studio].

Comments 10: [Did the author provide mitochondrial methods? I could not find the content].

Responses 10: [The supplementary material contains the requested mitochondrial methods].

Reviewer 3 Report

Comments and Suggestions for Authors

The paper is good and privides the reader with novel information on agronomic impacts concerning ryegrass production and certain changes in the morphology and nutrtional value of the plant. In favour of improving the value of the paper some modifications of that would be welcome.

The paper has no materials and methods chapter. Methodological information is scattered along the results passages. The readers would welcome a definite description of the trial and the observations and evaluations.

Also, there would be need for a brief summary of the production technologies of ryegrass on the given Peruvian cropsite (soil tillage, fertilization, plant protection, sowing, management, cutting etc).

In table 3 there is a good presentation of the certain morphological differentiation. I suggest the authors to put bigger than thumbnail photos since this evaluation is one of the most valuable result of the whole research.

In the discussion chapter a short the description of the forage preparation , handling and storage would be beneficial.

The English of the manuscript is good but woild need a thorough checking to track mistakes and typos.

Comments on the Quality of English Language

The paper has been written in an appropriate way. I suggested the authors a final proofreading and some changes in the structure of the manuscript.

Author Response

The article is good and provides the reader with novel information on the agronomic impacts related to ryegrass production and certain changes in the morphology and nutritional value of the plant. To improve the value of the article, some modifications would be welcome.

Comments 1: [The article does not contain a chapter on materials and methods. Methodological information is scattered in the results passages. Readers would appreciate a precise description of the trial and the observations and evaluations].

Response 1: [The article has taken into account the chapter on materials and methods, which appears between lines 355 and 432 of the original text].

Comments 2: [In addition, a brief summary of the technologies of ryegrass production in the Peruvian crop in question (soil tillage, fertilisation, plant protection, sowing, management, cutting, etc.) would be necessary].

Response 2: [The recommendations have been duly considered. The text provides a comprehensive and accurate overview of the production technologies employed in the cultivation of ryegrass in the Peruvian crop].

Comments 3: [The morphological differentiation is clearly presented in table 3. I suggest the authors to place larger pictures than the thumbnails, as this assessment is one of the most valuable results of the whole research].

Response 3: [The recommendations have been duly considered and the images have been enlarged and enhanced for the purpose of facilitating a more accurate and complete interpretation.]

Comments 4: [In the discussion chapter it would be beneficial to include a brief description of fodder preparation, handling and storage].

Response 4: [The description of the preparation, handling and storage of fodder has been taken into account in the discussion in accordance with the recommendations set out in the commentary].

Comments 5: The English of the manuscript is good, but would need a thorough revision for errors and typos. The article is adequately written. I suggested to the authors a final correction and some changes in the structure of the manuscript.

Response 5:  [As indicated in the commentary, the manuscript has been revised and subjected to rigorous editing for enhanced written English].

Round 2

Reviewer 1 Report

Comments and Suggestions for Authors

L.46-47 „there are no differences in the annual protein production at the cutting age of 60 and 45 days“ - I found this interesting statement only in Abstract. Please include it in the Results section.

L.169 not in most locations, but in all locations together. You didn't test the difference between the LM-43 and LM-58 at each location separately.

L.169-171 three times repetition

L.179-210 If the whole of Chapter 2.2 is only about LM-58 (which you imply in Response 6), then it needs to be clearly stated at the beginning of the chapter. Then also delete LM-43 from line 198.

L.205 Fig 1 correct carefully the units!

L.210 genotype

L.253 Table 6 - SD!, not DS

L.279 S1-S9

L.334 3.2.

L.338 Protein content 14.84% was at 30 days, not at 60 days (see Table 2). Please reword the entire sentence on lines 337-341 to make sense and match Table 2.

L.354-355 „in interaction with the locations evaluated“ - What do you mean? I assume that p=0.0036 is the value of the difference between LM-58 and LM-43 at all locations combined, not in interaction

L.440-441 Please indicate clearly that each genotype was grown in 2 exeprimental plots in each location.

L.540 But Table 9 does not contain any differences between LM-58 and LM-43!!.

Author Response

Comments 1: L.46-47 „there are no differences in the annual protein production at the cutting age of 60 and 45 days “- I found this interesting statement only in Abstract. Please include it in the Results section.

Response: [According to the suggested comment the inclusion has been made according to the requested section].

Comments 2: L.169 not in most locations, but in all locations together. You didn't test the difference between the LM-43 and LM-58 at each location separately.

Response: [The suggested comment has been taken into account]

Comments 3: [L.169-171 three times repetition].

Response: [The repetitions mentioned in comment have been deleted].

Comments 4: [L.179-210 If the whole of Chapter 2.2 is only about LM-58 (which you imply in Response 6), then it needs to be clearly stated at the beginning of the chapter. Then also delete LM-43 from line 198.].

Response: [The information has been modified according to the suggestions in the commentary].

Comments 5: [L.205 Fig 1 correct carefully the units!].  

Response: [The units have been duly corrected in accordance with the recommendations set forth in the commentary].

[Comments 6: [L.210 genotype].

Response: [The word mentioned in the comment has been modified].

Comments 7: [L.253 Table 6 - SD, not DS].

Response: [The acronyms recommended in the commentary have been modified].

Comments 8: [L.279 S1-S9].

Response: [Attached is what was requested by the comment].

Comments 9: [L.334 3.2.]

Response: [Corrections were made to the commentary on the above-mentioned line].

Comments 10: [L.338 Protein content 14.84% was at 30 days, not at 60 days (see Table 2). Please reword the entire sentence on lines 337-341 to make sense and match Table 2.].

Response: [The above lines were reformulated according to the specifications set out in the above-mentioned commentary].

Comments 11: [L.354-355 „in interaction with the locations evaluated “- What do you mean? I assume that p=0.0036 is the value of the difference between LM-58 and LM-43 at all locations combined, not in interaction].

Response: [The value of 0.036 refers to the combination of the frequency of cuts by location, taking into account that the two genotypes have not been evaluated in all locations, but only LM-58.]

Comments 12: [L.440-441 Please indicate clearly that each genotype was grown in 2 experimental plots in each location].

Response: [As requested in the commentary, the indications have been duly noted].

Comments 13: [L.540 But Table 9 does not contain any differences between LM-58 and LM-43!!].

Response: [Table 9 does not assess comparisons of the genotypes; rather, it refers to the general qualities to be assessed in both genotypes].

Round 3

Reviewer 1 Report

Comments and Suggestions for Authors

Dear authors,

I am sorry that you have only responded to a minor part of my comments, even though you write that you have taken them into account. I don't understand that. I certainly do not wish to do you any harm, but I must insist that you either take my comments into account in the manuscript or explain why they are wrong.

L.30-31 „there are no differences in the annual protein production at the cutting age of  60 and 45 days“ – where is the inclusion of this statement? I did not find it in the Results section…

L.248  not in most locations, but in all locations together. You didn't test the difference between the LM-43 and LM-58 at each location separately.

L.255-345 If the whole of Chapter 2.2 is only about LM-58 (which you imply in your previous Response ), then it needs to be clearly stated at the beginning of the chapter. Then also delete LM-43 from line 340

L.350 genotype

L.469 Table 6 – SD, not DS!!!

L.490 S1-S9

L.568 3.2.

L.572 Protein content 14.84% was at 30 days, not at 60 days (see Table 2). Please reword the entire sentence on lines 569-575 to make sense and match Table 2

L.605-607 Finally, it  should be noted that the genotype LM-58, in interaction of the frequency of cuts and the locations evaluated, obtained a significant value (p = 0.0036) for protein percentage.

L.700-701 Please indicate clearly that each genotype was grown in 2 experimental plots in each location

L.830 As you wrote, Table 9 does not assess comparisons of the genotypes. So please delete the word „differences“ from the line 830.

Author Response

I am sorry that you have only responded to a minor part of my comments, even though you write that you have taken them into account. I don't understand that. I certainly do not wish to do you any harm, but I must insist that you either take my comments into account in the manuscript or explain why they are wrong.

[Please accept our apologies; the incorrect format of the manuscript was uploaded in the previous submission. We look forward to your consideration and will proceed to upload the correct file].

Comments 1: [L.30-31 „there are no differences in the annual protein production at the cutting age of  60 and 45 days“ – where is the inclusion of this statement? I did not find it in the Results section…]

Response: [According to the suggested comment the inclusion has been made according to the requested section].

Comments 2: [L.248  not in most locations, but in all locations together. You didn't test the difference between the LM-43 and LM-58 at each location separately].

Response: [The suggested comment has been taken into account]

Comments 3: [L.255-345 If the whole of Chapter 2.2 is only about LM-58 (which you imply in your previous Response ), then it needs to be clearly stated at the beginning of the chapter. Then also delete LM-43 from line 340].

Response: [The information has been modified according to the suggestions in the commentary].

[Comments 4: [L.350 genotype].

Response: [The word mentioned in the comment has been modified].

Comments 5: [L.469 Table 6 - SD, not DS!!!].

Response: [The acronyms recommended in the commentary have been modified].

Comments 6: [L.490 S1-S9].

Response: [Attached is what was requested by the comment].

Comments 7: [L.568 3.2.]

Response: [Corrections were made to the commentary on the above-mentioned line].

Comments 8: [L.572 Protein content 14.84% was at 30 days, not at 60 days (see Table 2). Please reword the entire sentence on lines 569-575 to make sense and match Table 2].

Response: [The above lines were reformulated according to the specifications set out in the above-mentioned commentary].

Comments 9: [L.605-607 Finally, it  should be noted that the genotype LM-58, in interaction of the frequency of cuts and the locations evaluated, obtained a significant value (p = 0.0036) for protein percentage].

Response: [The value of 0.036 refers to the combination of the frequency of cuts by location, taking into account that the two genotypes have not been evaluated in all locations, but only LM-58.]

Comments 10: [L.700-701 Please indicate clearly that each genotype was grown in 2 experimental plots in each location].

Response: [As requested in the commentary, the indications have been duly noted].

Comments 11: [L.830 As you wrote, Table 9 does not assess comparisons of the genotypes. So please delete the word „differences“ from the line 830].

Response: [Table 9 does not assess comparisons of the genotypes; rather, it refers to the general qualities to be assessed in both genotypes].
